# A Pattern Dictionary Method for Anomaly Detection

**DOI:** 10.3390/e24081095

**Published:** 2022-08-09

**Authors:** Elyas Sabeti, Sehong Oh, Peter X. K. Song, Alfred O. Hero

**Affiliations:** 1Michigan Institute for Data Science, University of Michigan, Ann Arbor, MI 48109, USA; 2Department of Electrical Engineering and Computer Science, University of Michigan, Ann Arbor, MI 48109, USA; 3Department of Biostatistics, University of Michigan, Ann Arbor, MI 48109, USA

**Keywords:** pattern dictionary, atypicality, Lempel–Ziv algorithm, lossless compression, anomaly detection

## Abstract

In this paper, we propose a compression-based anomaly detection method for time series and sequence data using a pattern dictionary. The proposed method is capable of learning complex patterns in a training data sequence, using these learned patterns to detect potentially anomalous patterns in a test data sequence. The proposed pattern dictionary method uses a measure of complexity of the test sequence as an anomaly score that can be used to perform stand-alone anomaly detection. We also show that when combined with a universal source coder, the proposed pattern dictionary yields a powerful atypicality detector that is equally applicable to anomaly detection. The pattern dictionary-based atypicality detector uses an anomaly score defined as the difference between the complexity of the test sequence data encoded by the trained pattern dictionary (typical) encoder and the universal (atypical) encoder, respectively. We consider two complexity measures: the number of parsed phrases in the sequence, and the length of the encoded sequence (codelength). Specializing to a particular type of universal encoder, the Tree-Structured Lempel–Ziv (LZ78), we obtain a novel non-asymptotic upper bound, in terms of the Lambert W function, on the number of distinct phrases resulting from the LZ78 parser. This non-asymptotic bound determines the range of anomaly score. As a concrete application, we illustrate the pattern dictionary framework for constructing a baseline of health against which anomalous deviations can be detected.

## 1. Introduction

Anomaly detection and outlier detection are used for detecting data samples that are inconsistent with normal data samples. Early methods did not take the sequential structure of the data into consideration [1]. However, many real world applications involve data collected as a sequence or time series. In such data, anomalous samples are better characterized as subsequences of time series. Anomaly detection is a challenging task due to the uncertain nature of anomalies. Anomaly detection in time series and sequence data is particularly difficult since both length and occurrence frequency of potentially anomalous subsequences are unknown. Additionally, algorithmic computational complexity can be a challenge, especially for streaming data with large alphabet sizes.

In this paper, we propose a universal nonparametric model-free anomaly detection method for time series and sequence data based on a pattern dictionary (PD). Given training and test data sequences, a pattern dictionary is created from the sets of all the patterns in the training data. This dictionary is then used to sequentially parse and compress (in a lossless manner) the test data sequence. Subsequently, we interpret the number of parsed phrases or the codelength of the test data as anomaly scores. The smaller the number of parsed phrases or the shorter the compressed codelength of the test data, the more similarity between training and test data patterns. This sequential parsing and lossless compression procedure leads to detection of anomalous test sequences and their potential anomalous patterns (subsequences).

The proposed pattern dictionary method has the following properties: (i) it is nonparametric since it does not rely on a family of parametric distributions; (ii) it is universal in the sense that the detection criterion does not require any prior modeling of the anomalies or nominal data; (iii) it is non-Bayesian as the detection criterion is model-free; and (iv) as it depends on data compression, data discretization is required prior to building the dictionary. While the proposed pattern dictionary can be used as a stand-alone anomaly detection method (Pattern Dictionary for Detection (PDD)), we show how it can be utilized in the atypicality framework [2,3] for more general data discovery problems. This results in a method we call PDA (Pattern Dictionary based Atypicality), in which the proposed pattern dictionary is contrasted against a universal source coder which is the Tree-Structured Lempel–Ziv (LZ78) [4,5]. We use the LZ78 as the universal encoder since its compression procedure is similar to our proposed pattern dictionary, and it is (asymptotically) optimal [4,5].

The main contributions of this paper are as follows. First, we propose the pattern dictionary method for anomaly detection and characterize its properties. We show in Theorem 1 that using a multi-level dictionary that separates the patterns by their depth results in a shorter average indexing codelength in comparison to a uni-level dictionary that uses a uniform indexing approach. Second, we develop novel non-asymptotic lower and upper bounds of the LZ78 parser in Theorem 2 and further analyze its non-asymptotic properties. We demonstrate that the non-asymptotic upper bound on the number of distinct phrases resulting from the LZ78 parsing of an $\left|X\right|$-ary sequence of length *l* can be explicitly expressed by the Lambert W function [6]. To the best of our knowledge, such characterization has not previously appeared in the literature. Then, we show in Lemma 1 that the achieved non-asymptotic upper bound on the number of distinct phrases resulting from the LZ78 parsing converges to the optimal upper bound $\frac{l}{\log l}$ of the LZ78 parser as l→∞. Third, we show how the pattern dictionary and LZ78 can be used together in an atypicality detection framework. We demonstrate that the achieved non-asymptotic lower and upper bounds on both LZ78 and pattern dictionary determine the range of the anomaly score. Consequently, we show how these bounds can be used to analyze the effect of dictionary depth on the anomaly score. Furthermore, the bounds are used to set the anomaly detection threshold. Finally, we compare our proposed methods with the competing methods, including nearest neighbors-based similarity [7], threshold sequence time-delay embedding [8,9,10,11], and compression-based dissimilarity measure [12,13,14,15,16,16], that are designed for anomaly detection in sequence data and time series. We conclude our paper with an experiment that details how the proposed framework can be used to construct a baseline of health against which anomalous deviations are detected.

The paper is organized as follows. In Section 2, we briefly review the relevant literature in anomaly detection (readers who are familiar with anomaly detection can skip this section). Section 3 introduces the detection framework and the notation used in this paper. Section 4 presents our proposed pattern dictionary method and its properties. In Section 5, we show how the proposed pattern dictionary can be used in an atypicality framework alongside LZ78, and we analyze the non-asymptotic properties of the LZ78 parser. Section 6 presents experiments that illustrate the proposed pattern dictionary anomaly detection procedure. Finally, Section 7 concludes our paper.

## 2. Related Works

Anomaly detection has a vast literature. Anomaly detection procedures can be categorized into parametric and nonparametric methods. Parametric methods rely on a family of parametric distributions to model the normal data. The slippage problem [17], change detection [18,19,20,21], concept drift detection [19,20,21,22], minimax quickest change detection (MQCD) [23,24,25], and transient detection [26,27,28,29] are examples of parametric anomaly detection problems. The main difference between our proposed pattern dictionary method and the aforementioned techniques is that our method is a model-free nonparametric method. The main drawback of the parametric anomaly detection procedure is that it is difficult to accurately specify the parametric distribution for the data under investigation.

Nonparametric anomaly detection approaches do not assume any explicit parameterized model for the data distributions. An example is an adaptive nonparametric anomaly detection approach called geometric entropy minimization (GEM) [30,31] that is based on the minimal covering properties of *K*-point entropic graphs constructed on *N* training samples from a nominal probability distribution. The main difference between GEM-based methods and our proposed pattern dictionary is that former techniques are designed to detect outliers and cannot easily incorporate the temporal information regarding anomaly in a data stream. Another nonparametric detection method is sequential nonparametric testing that considers data as online stream and addresses the growing data storage problem by sequentially testing every new data samples [32,33]. A key difference between sequential nonparametric testing and our proposed pattern dictionary method is that our method is based on coding theory instead of statistical decision theory.

Information theory and universal source coding have been used previously in anomaly detection [34,35,36,37,38,39,40,41,42,43,44,45]. The detection criteria in these approaches are based on comparing metrics such as complexity or similarity distances that depend on entropy rate. An issue with these approaches is that there are many completely dissimilar sources with the same entropy rate, reducing outlier sensitivity. Another related problem is universal outlier detection [46,47]. In these works, different levels of knowledge about nominal and outlier distributions and number of outliers are incorporated. Unlike these methods, our proposed pattern dictionary approach does not require any prior knowledge about outliers and anomalies. In [48], a measure of empirical informational divergence between two individual sequences generated from two finite-order, finite-alphabet, stationary Markov processes is introduced and used for a simple universal classification. While the parsing procedure used in [48] is similar to the pattern dictionary used in this paper, there are important differences. The empirical measure proposed in [48] is a stand alone score function that is designed for two-class classification, while our measure is a direct byproduct of the LZ78 encoding algorithm designed for single-class classification, i.e., anomaly detection. In addition, the theoretical convergence of the empirical measure to the relative entropy between the class conditioned distributions, shown in [48], is only guaranteed when the sequences satisfy the finite-order Markov property, a condition that may be difficult to satisfy in practice. In [2,3], an information theoretic data discovery framework called *atypicality* has been introduced in which the detection criterion is based on a descriptive codelength comparison of an optimum encoder or a training-based fixed source coder, namely a data-dependent source coder introduced in [2]) with a universal source coder. In this paper, we show how our proposed pattern dictionary method can be used as a training-based fixed source coder in an atypicality framework.

Anomaly and outlier detection for time series has also been extensively studied [49]. Various time series modeling techniques such as regression [50], auto regression [51], auto regression moving average [52], auto regressive integrated moving average [53], support vector regression [54], and Kalman filters [55] have been used to detect anomalous observations by comparing the estimated residuals to a threshold. Many of these methods depend on a statistical assumption on the residuals, e.g., an assumption of Gaussian distribution, while the pattern dictionary method is model-free.

The proposed pattern dictionary method is closely related to the anomaly detection methods that are designed for sequence data. Many of these methods are focused on specific applications. For instance, detection of mutations in DNA sequences [7,56], detection of cyberattacks in computer network [57], and detection of irregular behaviors in online banking [58] are all application-specific examples of anomaly detection for discrete sequences. In the recent years, multiple sequence data anomaly detection methods have been developed specifically for graphs [59], dynamic networks [60], and social networks [61]. Chandola et al. [34] summarized many anomaly detection methods for discrete sequences and identified three general approaches to this problem. These anomaly detection formulations are unique in the way that anomalies are defined, but similar in their reliance on comparison between a test (sub)sequence and normal sequences in the training data. For example, kernel-based techniques such as nearest neighbor-based similarity (NNS) [7] are designed to detect anomalous sequences that are dissimilar to the training data. As another example, threshold sequence time-delay embedding (t-STIDE) [8,9,10,11] is established to detect anomalous sequences that contain subsequences with anomalous occurrence frequencies. The compression-based dissimilarity measure (CDM) is proposed for discord detection [12,13,14,15,16,16] to detect anomalous subsequences within a long sequence. Chandola et al. [34] also showed how various techniques developed for one problem formulation can be changed and applied to other problem formulations. While our pattern dictionary method shares similarity with NNS, CDM, and t-STIDE, our proposed method is generally applicable to any of the categories of anomaly detection identified in [34]. Furthermore, our detection criterion does not depend on the specific type of anomaly. Note that while CDM is also a compression-based method, its anomaly score is based on a dissimilarity measure that might fail to detect atypical subsequences [2]. For instance, using CDM method, a binary i.i.d. uniform training sequence is equally dissimilar to another binary i.i.d. uniform test sequence or to a test sequence drawn from some other distribution. In Section 6, the detection performance of our proposed pattern dictionary method is compared with NNS, CDM, t-STIDE, and the Ziv–Merhav method of [48].

It is worth mentioning that since the proposed pattern dictionary method is based on lossless source coding, it requires discretization of time series prior to deployment. In fact, many anomaly detection approaches require discretization of continuous data prior to applying inference techniques [62,63,64,65]. Note that discretization is also a requirement in other problem settings such as continuous optimization in genetic algorithms [66], image pattern recognition [67], and nonparametric histogram matching over codebooks in computer vision [68].

## 3. Framework and Notation

In the anomaly detection literature for sequence data and time series, the following three general formulations are considered [34]: (i) an entire test sequence is anomalous if it is notably different from normal training sequences; (ii) a subsequence within a long test sequence is anomalous if it is notably different from other subsequences in the same test sequence or the subsequences in a given training sequence; and (iii) a given test subsequence or pattern is anomalous if its occurrence frequency in a test sequence is notably different from its occurrence frequency in a normal training sequence. In this paper, we consider a unified formulation in which we determine if a (sub)sequence is anomalous with respect to a training sequence (or training sequence database) if any of the aforementioned three conditions are met. In other words, given a training sequence or a training sequence database, a test sequence is anomalous if it is significantly different from training sequences, or it contains a subsequence that is significantly different from subsequences in the training sequence, or it contains a subsequence whose occurrence frequency is significantly different from its occurrence frequency in the training data.

### Notation

We use *x* to denote a sequence and $x_{n}^{m}$ to denote a subsequence of *x*: $x_{n}^{m}=\left\{x_{i},i=n,n+1,\ldots ,m\right\}$, and $x^{l}$ represents a sequence of length *l*, i.e., $\left\{x_{n},n=1,\ldots ,l\right\}$. X denotes a finite set, and D represents a dictionary of subsequences. Throughout this paper:All logarithms are base 2 unless otherwise is indicated.In the encoding process, we always adhere to lossless compression and strict decodability at the decoder.While adhering to strict decodability, we only care about the codelength, not the codes themselves.

## 4. Pattern Dictionary: Design and Properties

Consider a long sequence, called the training data, $\left\{x_{n},n=1,\ldots ,L\right\}$ of length *L* drawn from a finite alphabet X. The goal is to *learn* the patterns (subsequences) of this sequence by creating a dictionary that contains all distinct patterns of maximum length (depth) $D_{max}\ll L$ that are embedded in the sequence. We call this dictionary a *pattern dictionary* D with the maximum depth $D_{max}$ and the set of observed patterns $S_{D}\left(x_{1}^{L}\right)$.

**Example** **1.**
*Suppose $D_{max}=2$, the alphabet is $X=\left\{A,B,C,D\right\}$ and the training sequence is x=ABACADABBACCADDABABACADAB. The set of patterns with depth $d\leq D_{max}$ in this sequence is $S_{D}\left(x\right)=\left\{A,B,C,D,AB,BA,AC,CA,AD,DA,BB,CC,DD\right\}$.*


Since the pattern dictionary is going to be used as a training-based fixed source coder (a data-dependent source coder as defined in [2]), an efficient structure for the pattern representation that minimizes the indexing codelength is of interest. The simplest approach is to consider all the patterns of length $1\leq d\leq D_{max}$ in one set $S_{D}$ and use a uniform indexing approach. This approach is called a *uni-level dictionary*. Another approach is to separate all the patterns by their depth (pattern length) and arrange them in $D_{max}$ sets $S_{D}^{\left(1\right)},S_{D}^{\left(2\right)},\ldots ,S_{D}^{\left(D_{\max}\right)}$, and define $S_{D}={⋃}_{d=1}^{D_{max}}S_{D}^{\left(d\right)}$, which we call a *multi-level dictionary*. In the following sections, we show that the latter results in a shorter average indexing codelength. It is worth mentioning that since a multi-level dictionary results in a depth-dependent indexing codelength, the average over the depth is considered. A relevant question is if the average of indexing codelength over all the patterns independent of depth should be used as an alternative. Since such pattern dictionaries are used to sequentially parse test data, patterns at smaller depth are more likely to be matched, even if they are anomalous. Thus, the average of indexing codelength over depth can better differentiate depth-dependent anomalies.

### 4.1. A Special Case

Suppose all the possible patterns of depth $d\leq D_{max}$ exist in the training sequence $\left\{x_{n},n=1,\ldots ,L\right\}$. That is, the cardinality of $S_{D}^{\left(d\right)}$ is $\left|S_{D}^{\left(d\right)}\right|={\left|X\right|}^{d}$ for $1\leq d\leq D_{max}$. Then, the total number of patterns is
$$\begin{array}{cc} \left|S_{D}\left(x_{1}^{L}\right)\right| & =\sum_{d=1}^{D_{max}}\left|S_{D}^{\left(d\right)}\left(x_{1}^{L}\right)\right|\\ \; & =\sum_{d=1}^{D_{max}}{\left|X\right|}^{d}\\ \; & =\frac{\left|X\right|\left({\left|X\right|}^{D_{max}}-1\right)}{\left|X\right|-1}. \end{array}$$Hence, a uni-level dictionary results in a uniform indexing codelength of
$$\begin{array}{cc} L^{uni} & =\log\left(\frac{\left|X\right|\left({\left|X\right|}^{D_{max}}-1\right)}{\left|X\right|-1}\right)\\ \; & \approx D_{max}\log\left(\left|X\right|\right). \end{array}$$On the other hand, a multi-level dictionary requires a two-stage description of index. The first stage is the index of the depth *d* (using $\log D_{max}$ bits), and the second stage is the index of the pattern among all the patterns with the same depth (using $d\log\left(\left|X\right|\right)$ bits). This two-stage description of the index leads to a non-uniform indexing of codelength: the minimum indexing codelength occurring for the patterns of depth d=1 equals to $L_{min}^{multi}=$$\log D_{max}+\log\left(\left|X\right|\right)$ bits, while the maximum indexing codelength occurring for the patterns of depth $d=D_{max}$ equals to $L_{max}^{multi}=$$\log D_{max}+D_{max}\log\left(\left|X\right|\right)$ bits. Thus, the average indexing codelength of a multi-level dictionary is given by
$$\begin{array}{cc} L^{multi} & =\frac{1}{D_{max}}\sum_{d=1}^{D_{max}}\left(\log D_{max}+d\log\left(\left|X\right|\right)\right)\\ \; & =\log D_{max}+\frac{\log\left(\left|X\right|\right)}{D_{max}}\sum_{d=1}^{D_{max}}d\\ \; & \approx \log D_{max}+\frac{1}{2}D_{max}\log\left(\left|X\right|\right). \end{array}$$Figure 1 and Figure 2 graphically compare the indexing codelength between a uni-level dictionary and a multi-level dictionary for a fixed alphabet size and a fixed $D_{max}$, respectively. As seen, the average indexing codelength of a multi-level dictionary results in a shorter indexing codelength.

### 4.2. The General Case

Given the training sequence $\left\{x_{n},n=1,\ldots ,L\right\}$, suppose there are $a_{d}=\left|S_{D}^{\left(d\right)}\right|\leq {\left|X\right|}^{d}$ patterns of depth $d\leq D_{max}$ ($a_{1}$ patterns of depth one, $a_{2}$ patterns of depth two, etc.). The following Theorem 1 shows that the average indexing codelength using a multi-level dictionary is always less than the indexing codelength of a uni-level dictionary.

**Theorem** **1.**
*Assume there are embedded $a_{d}=\left|S_{D}^{\left(d\right)}\right|\leq {\left|X\right|}^{d}$ patterns of depth $1\leq d\leq D_{max}$ in a training sequence of length $L\gg D_{max}$. Let $L^{uni}$ and $L^{multi}$ be the indexing codelength of a uni-level dictionary and the average indexing codelength of a multi-level dictionary, respectively. Then,*

*(1) $L^{multi}\leq$$L^{uni}$; and*

*(2) $\log\left(1+\frac{{\left(\sqrt{a_{D_{max}}}-\sqrt{a_{1}}\right)}^{2}}{D_{max}a_{D_{max}}}\right)\leq L^{uni}-L^{multi}\leq \log\left(1+w+\left(1-w\right)\frac{a_{D_{max}}}{a_{1}}-a_{1}^{w-1}a_{D_{max}}^{1-w}\right),$*

*where*

$$\begin{array}{cc} w & =\frac{\ln\left[\left(\frac{a_{D_{max}}}{a_{D_{max}}-a_{1}}\right)\ln\frac{a_{D_{max}}}{a_{1}}\right]}{\ln\frac{a_{D_{max}}}{a_{1}}}. \end{array}$$



**Proof.** Since $L\gg D_{max}$, clearly $0<a_{1}\leq a_{2}\leq \cdots \leq a_{D_{max}}$. Using a uni-level dictionary, the indexing codelength is
$$\begin{array}{cc} L^{uni} & =\log\left(\sum_{d=1}^{D_{max}}a_{d}\right)\\ \; & =\log D_{max}+\log A_{D_{max}}, \end{array}$$
where $A_{D_{max}}≜\left(a_{1}+a_{2}+\cdots +a_{D_{max}}\right)/D_{max}$ is the arithmetic mean of $a_{1},a_{2},\ldots ,a_{D_{max}}$. Using a multi-level dictionary the average indexing codelength is
$$\begin{array}{cc} L^{multi} & =\frac{1}{D_{max}}\sum_{d=1}^{D_{max}}\left(\log D_{max}+\log a_{d}\right)\\ \; & =\log D_{max}+\log G_{D_{max}}, \end{array}$$
where $G_{D_{max}}≜{\left(\prod_{d=1}^{D_{max}}a_{d}\right)}^{1/D_{max}}$ is the geometric mean of $a_{1},a_{2},\ldots ,a_{D_{max}}$. Hence, the comparison between $L^{uni}$ and $L^{multi}$ comes down to comparing the arithmetic mean and the geometric mean of $a_{1},a_{2},\ldots ,a_{D_{max}}$. Thus, $A_{D_{max}}\geq G_{D_{max}}$, which established the first part of the theorem. For the second part of the theorem, we use lower and upper bounds on $A_{D_{max}}-G_{D_{max}}$ derived in [69]
$$\begin{array}{cc} \frac{{\left(\sqrt{a_{D_{max}}}-\sqrt{a_{1}}\right)}^{2}}{D_{max}} & \leq A_{D_{max}}-G_{D_{max}}\leq \\ \; & \left[wa_{1}+\left(1-w\right)a_{D_{max}}-a_{1}^{w}a_{D_{max}}^{1-w}\right], \end{array}$$
where $w=\frac{\ln\left[\left(a_{D_{max}}/\left(a_{D_{max}}-a_{1}\right)\right)\ln\left(a_{D_{max}}/a_{1}\right)\right]}{\ln\left(a_{D_{max}}/a_{1}\right)}$. Since $a_{1}\leq G_{D_{max}}\leq a_{D_{max}}$ and $L^{uni}-L^{multi}=\log\frac{A_{D_{max}}}{G_{D_{max}}}$, the proof is complete. □

Theorem 1 shows that a multi-level dictionary gives shorter average indexing codelength than a uni-level dictionary. $\log D_{max}+\log a_{d}$ is the indexing codelength for patterns of depth *d*, where $a_{d}$ is the total number of observed patterns of the depth *d*. In order to reduce the indexing codelength even further, the patterns of the same length in each set $S_{D}^{\left(d\right)}$ can be ordered according to their relative frequency (empirical probability) in the training sequence. This allows Huffman or Shannon–Fano–Elias source coding [4] to be used to assign prefix codes to patterns in each set $S_{D}^{\left(d\right)}$ separately. In this case, for any pattern $x_{1}^{d}\in S_{D}^{\left(d\right)}$, the indexing codelength becomes
(1)$$\begin{array}{cc} L^{multi} & \left(x_{1}^{d}\right)=\log D_{max}+L_{D}^{\left(d\right)}\left(x_{1}^{d}\right), \end{array}$$
where $L_{D}^{\left(d\right)}\left(x_{1}^{d}\right)$ is the codelength assigned to the pattern $x_{1}^{d}$ based on its empirical probability using a Huffman or Shannon–Fano–Elias encoder. If such encoders are used, the codelength (Equation 1) is optimal ([4] Theorem 5.8.1). Since the whole purpose of creating a pattern dictionary is to learn the patterns in the training data, assigning the shorter codelength to the more frequent patterns and assigning longer codelength to the less frequent patterns in any pattern set $S_{D}^{\left(d\right)}$ will improve the efficiency of the coded representation.

**Example** **2.**
*Suppose the alphabet is $X=\left\{A,B,C,D\right\}$ and the training sequence is x=ABACADABBACCADDABABACADAB. Table 1 shows the dictionary with $D_{max}=3$ created by the patterns inside the training sequence, and the codelength assigned for each pattern using Huffman coding.*


### 4.3. Pattern Dictionary for Detection (PDD)

Suppose we want to sequentially compress a test sequence $x_{1}^{l}=\left\{x_{n},n=1,\ldots ,l\right\}$ using a trained pattern dictionary D with maximum depth $D_{max}<l$. The encoder parses the test sequence $x_{1}^{l}$ into *c* phrases, $x_{v_{1}}^{v_{2}-1},x_{v_{2}}^{v_{3}-1},\ldots ,x_{v_{c}}^{l}$ where $v_{i}$ is the index of the start of the *i*th phrase, and each phrase $x_{v_{i}}^{v_{i+1}-1}$ is a pattern in the pattern dictionary D. Let $S_{D}\left(x_{1}^{l}\right)=\left\{x_{v_{1}}^{v_{2}-1},x_{v_{2}}^{v_{3}-1},\ldots ,x_{v_{c}}^{l}\right\}$ denote the set of the parsed phrases using pattern dictionary D. The parsing process begins with setting $v_{1}=1$ and finding the largest $v_{2}\leq D_{max}$ and $v_{2}\leq l$ such that $x_{v_{1}}^{v_{2}-1}\in D$ but $x_{v_{1}}^{v_{2}}\notin D$. This results in the first phrase $x_{1}^{v_{2}-1}$. Similarly, the same procedure is performed in order to find the largest $v_{3}\leq D_{max}$ and $v_{3}\leq l$ such that $x_{v_{2}}^{v_{3}-1}\in D$ but $x_{v_{2}}^{v_{3}}\notin D$. This type of cross-parsing was first introduced in [48] in order to estimate an empirical relative entropy between two individual sequences that are independent realizations of two finite-order, finite-alphabet and stationary Markov processes. Here, we do not impose such an assumption on the sources generating the sequences. Algorithm 1 summarizes the procedure of the proposed pattern dictionary (PD) parser. After parsing the whole test sequence $x_{1}^{l}$ into *c* phrases, $x_{v_{1}}^{v_{2}-1},x_{v_{2}}^{v_{3}-1},\ldots ,x_{v_{c}}^{l}$, the codelength will be
(2)$$\begin{array}{cc} L\left(x_{1}^{l}\right) & =\sum_{i=1}^{c}L_{D}\left(x_{v_{i}}^{v_{i+1}-1}\right)+c\log D_{max}. \end{array}$$

**Algorithm 1** Pattern Dictionary (PD) Parser
**Require**: Pattern Dictionary D, Test Sequence $x_{1}^{l}$1:Set c=1, $v_{c}=1$, d=12:
**while**

$\;v_{c}+d-1<l\;$

**do**
3:    **if** $x_{v_{c}}^{v_{c}+d-1}\in S_{D}^{\left(d\right)}$ **then**4:        **if** $d+1\leq D_{max}$ **then**5:           d=d+16:        **else**7:           $v_{c+1}=v_{c}+d$8:           c=c+19:           d=110:    **else**11:        $v_{c+1}=v_{c}+d-1$12:        c=c+113:        d=114:
**return**

$\;x_{v_{1}}^{v_{2}-1},x_{v_{2}}^{v_{3}-1},\ldots ,x_{v_{c}}^{l}$




For detection purposes, on a test sequence $x_{1}^{l}$, either the number of parsed phrases or the codelength can be used as anomaly scores with respect to the trained pattern dictionary D. In other words, for any test sequence $x_{1}^{l}$ and given a pattern dictionary, if the number of parsed phrases $\left|S_{D}\left(x_{1}^{l}\right)\right|$ or the codelength $L\left(x_{1}^{l}\right)$ in Equation (Equation 2) are greater than a certain threshold, then $x_{1}^{l}$ is declared to be anomalous. While the proposed pattern dictionary technique can be used as a stand-alone anomaly detection technique, below we show how it can be used for atypicality detection [2,3] as a training-based fixed source coder (data-dependent encoder).

## 5. Pattern Dictionary-Based Atypicality (PDA)

In [2,3], an *atypicality framework* was introduced as a data discovery and anomaly detection framework that is based on a central definition: “a sequence (or subsequence) is atypical if it can be described (coded) with fewer bits in itself rather than using the (optimum) code for typical sequences”. In this framework, detection is based on the comparison of a lossless descriptive codelength between an optimum encoder (if the typical model is known) or a training-based fixed source coder (if the typical model is unknown, but training data are available) and a universal source coder in order to detect atypical subsequences in the data [2,3]. In this section, we apply our proposed pattern dictionary as a training-based fixed source coder (typical encoder) in an atypicality framework. We call it pattern dictionary-based atypicality (PDA) method.

The pattern dictionary-based source coder can be considered as a generalization of the Context Tree [70,71,72] based fixed source coder that was used in [2] for discrete data. The universal source coder (atypical encoder) used here is the Tree-Structured Lempel–Ziv (LZ78) [4,5]. The primary reason for choosing LZ78 as the universal encoder is that its sequential parsing procedure is similar to the proposed pattern dictionary described in Section 4, and it is (asymptotically) optimal [4,5]. One might ask why do we even need to compare descriptive codelengths of a training-based (or optimum) encoder with a universal encoder for data discovery purposes when, as alluded to in the end of last section, a training-based fixed source coder can be a stand-alone anomaly detector. The necessity of such concurrent comparison is articulated in [2]. In fact, such a codelength comparison enables the atypicality framework to go beyond the detection of anomalies and outliers, extending to the detection of *rare* parts of data that might have a data structure of interest to the practitioner.

We give an example to provide further intuition for why anomaly detection can benefit from our framework that compares the outputs of a typical encoder and an atypical encoder. Consider an i.i.d. binary sequence of length *L* with $P\left(X=1\right)=p$ in which there is embedded an anomalous subsequence of length l≪L with $P\left(X=1\right)=\hat{p}\neq p$ that we would like to detect. If $p=\frac{1}{2}$ and $\hat{p}=1$, the typical encoder cannot catch the anomaly while the atypical encoder can. On the other hand, if $p=\frac{1}{3}$ and $\hat{p}=\frac{2}{3}$, the typical encoder identifies the anomaly while an atypical encoder fails to do so (since the entropy for $p=\frac{1}{3}$ and $\hat{p}=\frac{2}{3}$ is the same). Note that in both cases, our framework would catch the anomaly since it uses the difference between the descriptive codelengths of these two encoders.

Recall that in Section 4, we supposed that a test sequence $x_{1}^{l}$ has been parsed using a trained pattern dictionary D with maximum depth $D_{max}<l$. This parsing results in $\left|S_{D}\left(x_{1}^{l}\right)\right|$ parsed phrases. Using Equation (Equation 2), the typical codelength of the sequence $x_{1}^{l}$ is given by
$$\begin{array}{cc} L_{T}\left(x_{1}^{l}\right) & =\sum_{y\in S_{D}\left(x_{1}^{l}\right)}L_{D}\left(y\right)+\left|S_{D}\left(x_{1}^{l}\right)\right|\log D_{max}. \end{array}$$For the atypical encoder, the LZ78 algorithm results in a distinct parsing of the test sequence $x_{1}^{l}$. Let $S_{LZ}\left(x_{1}^{l}\right)$ denote the set of parsed phrases in the LZ78 parsing of $x_{1}^{l}$. As such, the resulting atypical codelength is [4,5]
$$\begin{array}{cc} L_{A}\left(x_{1}^{l}\right) & =\left|S_{LZ}\left(x_{1}^{l}\right)\right|\left[\log\left|S_{LZ}\left(x_{1}^{l}\right)\right|+1\right]. \end{array}$$

Since $L\left(x_{1}^{l}\right)$ using both LZ78 and the pattern dictionary depends on the number of parsed phrases, we investigate the possible range and properties of $\left|S_{D}\left(x_{1}^{l}\right)\right|-\left|S_{LZ}\left(x_{1}^{l}\right)\right|$. While the LZ78 encoder is a well-known compression method which is asymptotically optimal [4,5], its non-asymptotic behavior is not well understood. In the next section, we establish a novel non-asymptotic property of an LZ78 parser, and then compare it with the pattern dictionary parser.

### 5.1. Lempel–Ziv Parser

We start this section with a theorem that establishes the non-asymptotic lower and upper bounds on the number of distinct phrases in a sequence parsed by LZ78.

**Theorem** **2.**
*The number of distinct phrases $c\left(l\right)$ resulting from LZ78 parsing of an $\left|X\right|$-ary sequence $x_{1}^{l}=\left\{x_{n},n=1,\ldots ,l\right\}$ satisfies*

$$\begin{array}{cc} \frac{1}{2}\left(\sqrt{8l+1}-1\right)\leq c\left(l\right) & \leq \frac{l\ln\left|X\right|}{W\left(\frac{\beta }{\alpha }{\left|X\right|}^{\frac{\alpha +1}{-\alpha }}\ln\left|X\right|\right)}, \end{array}$$

*where $\alpha =\left|X\right|-1$, $\beta ={\left(\left|X\right|-1\right)}^{2}l-\left|X\right|$, and $W\left(.\right)$ is the Lambert W function [6].*


**Proof.** First, we establish the upper bound. Note that the number of parsed distinct phrases $c\left(l\right)$ is maximized when all the phrases are as short as possible. Define $M≜\left|X\right|$ and let $l_{k}$ be the sum of the lengths of all distinct strings of length less than or equal to *k*. Then,
$$\begin{array}{cc} l_{k} & =\sum_{j=1}^{k}jM^{j}=\frac{1}{{\left(M-1\right)}^{2}}\left[\left\{\left(M-1\right)k-1\right\}M^{k+1}+M\right]. \end{array}$$Since $l=l_{k}$ occurs when all the phrases are of length ≤k,
$$\begin{array}{cc} c\left(l_{k}\right) & \leq \sum_{j=1}^{k}M^{j}=\frac{M\left(M^{k}-1\right)}{M-1}<\frac{M^{k+1}}{M-1}\leq \frac{l_{k}}{k-\frac{1}{M-1}}. \end{array}$$If $l_{k}\leq l<l_{k+1}$, we write $l=l_{k}+▵$ where
$$\begin{array}{cc} ▵<l_{k+1}-l_{k} & =\left(Mk+M-1-k\right)\frac{M^{k+1}}{M-1}\\ \; & =\left(k+1\right)\frac{M^{k+1}}{M-1}. \end{array}$$We conclude that the parsing ends up with $c\left(l_{k}\right)$ phrases of length ≤k and $\frac{l-l_{k}}{k+1}$ phrases of length k+1. Therefore,
(3)$$\begin{array}{cc} c\left(l\right) & \leq c\left(l_{k}\right)+\frac{l-l_{k}}{k+1}\leq \frac{l_{k}}{k-\frac{1}{M-1}}+\frac{▵}{k+1}\\ \; & \leq \frac{l_{k}+▵}{k-\frac{1}{M-1}}=\frac{l}{k-\frac{1}{M-1}}. \end{array}$$We now bound the size of *k* for a given sequence of length *l* by setting $l=l_{k}$. Define α≜M−1 and $\beta ≜{\left(M-1\right)}^{2}l-M$. Then,
$$\begin{array}{cc} \; & \frac{1}{{\left(M-1\right)}^{2}}\left[\left(\left(M-1\right)k-1\right)M^{k+1}+M\right]=l\\ ⟺ & \left(\left(M-1\right)k-1\right)M^{k+1}={\left(M-1\right)}^{2}l-M\\ ⟺ & \left(\alpha k-1\right)M^{k+1}=\beta \\ ⟺ & \;\hat{k}M^{\left(\hat{k}+1\right)/\alpha +1}=\beta \\ ⟺ & \;\hat{k}\frac{\ln M}{\alpha }\exp\left(\hat{k}\frac{\ln M}{\alpha }\right)=\frac{\beta }{\alpha }M^{-1-1/\alpha }\ln M. \end{array}$$
where $\hat{k}=\alpha k-1$. The last equation can be solved using the Lambert W function [6]. Since all the involved numbers are real and for M>1 and l≥2, we have $\frac{\beta }{\alpha }M^{-1-1/\alpha }\ln M\geq 0>-\frac{1}{e}$, it follows that
$$\begin{array}{cc} \; & \hat{k}\frac{\ln M}{\alpha }=W\left(\frac{\beta }{\alpha }M^{-1-1/\alpha }\ln M\right)\\ ⟺ & k=\frac{\alpha W\left(\frac{\beta }{\alpha }M^{-1-1/\alpha }\ln M\right)+\ln M}{\alpha \ln M}, \end{array}$$
where $W\left(.\right)$ is the Lambert W function. Using equation (Equation 3), we write
$$\begin{array}{cc} c\left(l\right) & \leq \frac{l}{k-\frac{1}{\alpha }}=\frac{l\ln M}{W\left(\frac{\beta }{\alpha }M^{-1-1/\alpha }\ln M\right)}. \end{array}$$To prove the lower bound, note that the number of parsed distinct phrases $c\left(l\right)$ is minimized when the sequence of length *l* consists of only one symbol that repeats. Let $\tilde{l_{k}}$ be the sum of the lengths of all such distinct strings of length less than or equal to *k*. Then,
$$\begin{array}{cc} \tilde{l_{k}} & =\sum_{j=1}^{k}j=\frac{k\left(k+1\right)}{2}. \end{array}$$Thus, given a sequence of length *l* by enforcing $l=\frac{k\left(k+1\right)}{2}$, we obtain the lower bound. □

Figure 3 illustrates the lower and upper bounds established in Theorem 2 against the sequence length for various alphabet sizes. Note that the lower bound on the number of distinct phrases is independent of the alphabet size.

While numerical experiments are not a substitute for the mathematical proof of Theorem 2 provided above, the reader may find it useful to understand the theorem in terms of a simple example. In Figure 4, Figure 5 and Figure 6, we compare the theoretical bound with numerical results of simulation for binary i.i.d. sequences. In these experiments, for each value of P(X=1), a thousand binary sequences are generated; then, the number of distinct phrases resulting from LZ78 parsing of each sequence is calculated, and hence, the average, minimum, and maximum of these counts are found and represented by error bars.

Next, we verify the convergence of the non-asymptotic upper bound achieved in Theorem 2 to the asymptotic upper bound of the LZ78 parser. Using a lower bound on Lambert W function $\ln x-\ln\left(\ln x\right)\leq W\left(x\right)$ [73], we write
$$\begin{array}{cc} W\left(\frac{\beta }{\alpha }\frac{\ln M}{M^{1+1/\alpha }}\right) & =W\left(\left(\left(M-1\right)l-\frac{M}{M-1}\right)\frac{\ln M}{M^{\frac{M}{M-1}}}\right)\\ \; & \approx W\left(c_{M}l\ln M\right)\\ \; & \geq \ln\frac{c_{M}l\ln M}{\ln\left(c_{M}l\ln M\right)}\\ \; & =\ln\frac{c_{M}l}{\log\left(c_{M}l\ln M\right)}, \end{array}$$where the logarithm is base $M=\left|X\right|$ and $c_{M}=\frac{M-1}{M^{M/\left(M-1\right)}}$. Hence, we can further simplify the asymptotic upper bound of $c\left(l\right)$ as follows
$$\begin{array}{cc} c\left(l\right) & \leq \frac{l\ln M}{W\left(\frac{\beta }{\alpha }M^{-1-1/\alpha }\ln M\right)}\\ \; & \leq \frac{l\ln M}{\ln\frac{c_{M}l}{\log\left(c_{M}l\ln M\right)}}\\ \; & =\frac{l}{\log\frac{c_{M}l}{\log\left(c_{M}l\ln M\right)}}\\ \; & =\frac{l}{\log l+\log c_{M}-\log\log\left(c_{M}l\ln M\right)}\\ \; & =\frac{l}{\left(1-\frac{\log\log l+\hat{c_{M}}}{\log l}\right)\log l}, \end{array}$$
where $\hat{c_{M}}=\log c_{M}-\log\log\left(c_{M}\ln M\right)$. Therefore, as l→∞, we have $c\left(l\right)\leq \frac{l}{\log l}$. This is consistent with the binary case M=2 proved in ([4] Lemma 13.5.3) or [5]. The following Lemma extends the result of ([4] Lemma 13.5.3) to $\left|X\right|$-ary case.

**Lemma** **1.**
*The number of distinct phrases $c\left(l\right)$ resulting from LZ78-parsing of an $\left|X\right|$-ary sequence $x_{1}^{l}=\left\{x_{n},n=1,\ldots ,l\right\}$ satisfies*

$$\begin{array}{cc} c\left(l\right) & \leq \frac{l}{\left(1-\epsilon_{l}\right)\log l}, \end{array}$$


*where the logarithm is base $\left|X\right|$ and $\epsilon_{l}=\min\left\{1,\frac{\log\log l-\log\left(\left|X\right|-1\right)+\frac{3\left|X\right|-2}{\left|X\right|-1}}{\log l}\right\}\to 0$ as l→∞.*


**Proof.** The proof is similar to the proof in ([4] Lemma 13.5.3) or ([74] Theorem 2). Let $M≜\left|X\right|$. In Theorem 2, we defined $l_{k}$ as the sum of the lengths of all distinct strings of length less than or equal to *k*, and we showed that for any given *l* such that $l_{k}\leq l<l_{k+1}$, we have $c\left(l\right)\leq c\left(l_{k}\right)+\frac{l-l_{k}}{k+1}\leq \frac{l}{k-\frac{1}{M-1}}$. Next, we bound the size of *k*. As such, we have $l\geq l_{k}\geq M^{k}$ or, equivalently, k≤logl where the logarithm is base M. Additionally,
$$\begin{array}{cc} l\leq l_{k+1} & =\left(k+1-\frac{1}{M-1}\right)\frac{M^{k+2}}{M-1}+\frac{M}{{\left(M-1\right)}^{2}}\\ \; & =\left(\frac{k}{M-1}+\frac{M-2}{{\left(M-1\right)}^{2}}\right)M^{k+2}+\frac{M}{{\left(M-1\right)}^{2}}\\ \; & \leq \frac{k+2}{M-1}M^{k+2}\leq \frac{\log l+2}{M-1}M^{k+2}, \end{array}$$
therefore, $k+2\geq \log\frac{\left(M-1\right)l}{\log l+2}$. Equivalently, for $l\geq M^{2}$,
$$\begin{array}{cc} k-\frac{1}{M-1} & \geq \log l-\log\left(\log l+2\right)+\log\left(M-1\right)-2-\frac{1}{M-1}\\ \; & =\left(1-\frac{\log\left(\log l+2\right)-\log\left(M-1\right)+\frac{2M-1}{M-1}}{\log l}\right)\log l\\ \; & \geq \left(1-\frac{\log\left(2\log l\right)-\log\left(M-1\right)+\frac{2M-1}{M-1}}{\log l}\right)\log l\\ \; & =\left(1-\frac{\log\log l-\log\left(M-1\right)+\frac{3M-2}{M-1}}{\log l}\right)\log l\\ \; & =\left(1-\epsilon_{l}\right)\log l, \end{array}$$
where $\epsilon_{l}=\min\left\{1,\frac{\log\log l-\log\left(M-1\right)+\frac{3M-2}{M-1}}{\log l}\right\}$. □

Next, we analyze the properties of the number of distinct phrases $c\left(l\right)$ resulting from LZ78-parsing of an $\left|X\right|$-ary sequence $x_{1}^{l}=\left\{x_{n},n=1,\ldots ,l\right\}$ when *l* is fixed. The error bar representation in Figure 4 shows the variation of $c\left(l\right)$ when *l* is fixed. A possible explanation for such variations is that the statistical distribution of the pseudorandomly generated data are different from the theoretical distribution of the generating source. To elucidate this possibility, we enforce the exact matching of the source probability mass function and the empirical probability mass function of the generated data. Figure 5 represents the number of distinct phrases $c\left(l\right)$ resulting from LZ78-parsing of a binary sequence of fixed length where the characteristic of the generating source and the generated data matches. As seen, there is still some variation around the average value of $c\left(l\right)$. We can specify a distribution-dependent bound on $c\left(l\right)$ when both *l* and the distribution of the source are fixed.

In ([75] Theorem 1), for sequences generated from a memoryless source, $c\left(l\right)$ is assumed to be a random variable with the following mean and variance:(4)$$\begin{array}{cc} E\left(c\left(l\right)\right) & ∼\frac{hl}{\log l},\\ \mathrm{Var}\left(c\left(l\right)\right) & ∼\frac{\left(h_{2}-h^{2}\right)l}{\log^{2}l}, \end{array}$$
where $h=-\sum_{a\in X}p_{a}\log p_{a}$ is the entropy rate, and $h_{2}=\sum_{a\in X}p_{a}\log^{2}p_{a}$ with $p_{a}$ being the probability of symbol a∈X. Note that the approximations (Equation 4) are asymptotic as l→∞. Below, we obtain a finite sample characterization of $c\left(l\right)$.

Consider an $\left|X\right|$-ary sequence $x_{1}^{l}=\left\{x_{n},n=1,\ldots ,l\right\}$ with fixed length *l* generated from a source with the probability mass function $p\left(x\right)$. Here, the notations $x_{1}^{l}$ and $x^{l}$ are used interchangeably. Let $c\left(l,p\right)$ denote the number of distinct phrases resulting from LZ78-parsing of the sequence $x_{1}^{l}$ of length *l* and the generating probability mass function is defined by $p\left(x\right)$. In order to find a distribution-dependent bound on the number of distinct phrases in LZ78-based parsing of $x_{1}^{l}$, we note that since the generating distribution is not necessarily uniform, all the strings $x^{n}$ for n<l≪∞ do not necessarily appear as parsed phrases. For instance, consider the binary case with $P\left(X=1\right)=0.9$. Then, it is very unlikely to have a string with multiple consecutive zeros in any parsing of a realization of the finite sequence $x^{l}$. As such, using the Asymptotic Equipartition Properties (AEP) ([4] Chapter 3) or Non-asymptotic Equipartition Properties (NEP) [76], we define the *typical set* $A_{\epsilon }^{\left(n\right)}$ with respect to $p\left(x\right)$ as the set of subsequences $x^{n}\in X^{n}$ of $x_{1}^{l}$ with the property
$$\begin{array}{cc} 2^{-n\left(h+\epsilon \right)}\leq & p\left(x^{n}\right)\leq 2^{-n\left(h-\epsilon \right)}, \end{array}$$
where *h* is the entropy. Then, we have
$$\begin{array}{cc} 1 & =\sum_{x^{n}\in X^{n}}p\left(x^{n}\right)\geq \sum_{x^{n}\in A_{\epsilon }^{\left(n\right)}}p\left(x^{n}\right)\geq \left|A_{\epsilon }^{\left(n\right)}\right|2^{-n\left(h+\epsilon \right)}, \end{array}$$
therefore, $\left|A_{\epsilon }^{\left(n\right)}\right|\leq 2^{n\left(h+\epsilon \right)}.$ Let $l_{k}$ be the sum of the lengths of all the distinct strings $x^{n}$ in the set $\left|A_{\epsilon }^{\left(n\right)}\right|$ of length less than or equal to *k*. We write,
$$\begin{array}{cc} l_{k} & =\sum_{n=1}^{k}n\left|A_{\epsilon }^{\left(n\right)}\right|\\ \; & \leq \sum_{n=1}^{k}n2^{n\left(h+\epsilon \right)}\\ \; & =\frac{1}{{\left(m-1\right)}^{2}}\left[\left(\left(m-1\right)k-1\right)m^{k+1}+m\right], \end{array}$$
where $m≜2^{h+\epsilon }$. Therefore, $l=\frac{1}{{\left(m-1\right)}^{2}}\left[\left(\left(m-1\right)k-1\right)m^{k+1}+m\right]$ can be solved for *k* which leads into an upper bound for $c\left(l,p\right)$ as follows
$$\begin{array}{cc} k= & \frac{\alpha W\left(\frac{\beta }{\alpha }m^{-1-1/\alpha }\ln m\right)+\ln m}{\alpha \ln m}\\ c\left(l,p\right) & \leq \sum_{n=1}^{k}\left|A_{\epsilon }^{\left(n\right)}\right|=\frac{m\left(m^{k}-1\right)}{m-1}\\ \; & =\frac{2^{k\left(h+\epsilon \right)}-1}{1-2^{-h-\epsilon }}, \end{array}$$
where α=m−1 and $\beta ={\left(m-1\right)}^{2}l-m$. Therefore, the dependency of the $c\left(l,p\right)$ upper bound on the distribution is only through the entropy. Figure 6 depicts the upper bound on $c\left(l,p\right)$ for ϵ=0.1.

### 5.2. Pattern Dictionary Parser versus LZ78 Parser

Given an $\left|X\right|$-ary sequence $x_{1}^{l}=\left\{x_{n},n=1,\ldots ,l\right\}$, let $c_{T}\left(l\right)$ be the number of parsed phrases of $x_{1}^{l}$ when the typical encoder (pattern dictionary with $D_{max}$) is used, and $c_{A}\left(l\right)$ be the number of parsed phrases of $x_{1}^{l}$ when the atypical encoder (LZ78) is used. Clearly, $\frac{l}{D_{max}}\leq c_{T}\left(l\right)\leq l$ where the lower bound is achieved when $S_{D}\left(x_{1}^{l}\right)=\left\{x_{v_{1}}^{v_{2}-1},x_{v_{2}}^{v_{3}-1},\ldots ,x_{v_{c}}^{l}\right\}$, and each $x_{v_{i}}^{v_{i}-1}\in S_{D}^{\left(D_{\max}\right)}$, namely $x_{v_{i}}^{v_{i}-1}$ is of length $D_{max}$ and exists in the dictionary. The upper bound is achieved when $S_{D}\left(x_{1}^{l}\right)=\left\{x_{1},x_{2},\ldots ,x_{l}\right\}$ where each $x_{n}\in S_{D}^{\left(1\right)}$. Using the result of Theorem 2 and a lower bound on the Lambert W function, $\ln x-\ln\left(\ln x\right)\leq W\left(x\right)$ [73], we have
(5)$$\begin{array}{cc} \; & \frac{l}{D_{max}}\left(1-\frac{D_{max}}{\log\frac{l}{\log\left(l\ln\left|X\right|\right)}}\right)\leq c_{T}\left(l\right)-c_{A}\left(l\right)\\ \; & \;\;\;\;\;\leq l\left(1-\frac{\sqrt{8l+1}-1}{2l}\right). \end{array}$$The above bounds have asymptotic and non-asymptotic implications. The asymptotic analysis of the bounds in (Equation 5) suggests that as l→∞, for a dictionary with fixed $D_{max}$, we have $\frac{l}{D_{max}}\leq c_{T}\left(l\right)-c_{A}\left(l\right)\leq l$. This inequality implies the asymptotic dominance of the parser using a typical encoder. This is to be expected due to the asymptotic optimality of LZ78. However, the above inequality also implies a more interesting result: if $D_{max}>\log\frac{l}{\log\left(l\ln\left|X\right|\right)}$ as l→∞, then $c_{T}\left(l\right)$ can be smaller than $c_{A}\left(l\right)$. The non-asymptotic behavior of the bounds in (Equation 5) is more relevant to the anomaly detection problem. These bounds suggest that for a fixed *l* and $\left|X\right|$, increasing $D_{max}$ has a vanishing effect on the possible range of the anomaly score. Additionally, the achieved bounds on $c_{T}\left(l\right)-c_{A}\left(l\right)$ provide the range of values of the anomaly score. This facilitates the search for a data-dependent threshold for anomaly detection, as the search can be restricted to this range.

### 5.3. Atypicality Criterion for Detection of Anomalous
Subsequences

Consider the problem of finding the atypical (anomalous) subsequences of a long sequence with respect to a trained pattern dictionary D. Suppose we are looking for an infrequent anomalous subsequence $x_{n}^{n+l-1}=\left\{x_{n},n=n,\ldots ,n+l-1\right\}$ embedded in a test sequence $\left\{x_{n},n=1,\ldots ,L\right\}$ from the finite alphabet X. Using Equation (Equation 2), the typical codelength of the subsequence $x_{n}^{n+l-1}$ is
$$\begin{array}{cc} L_{T}\left(x_{n}^{n+l-1}\right) & =\sum_{y\in S_{D}\left(x_{n}^{n+l-1}\right)}L_{D}\left(y\right)+\left|S_{D}\left(x_{n}^{n+l-1}\right)\right|\log D_{max}, \end{array}$$
while using LZ78, the atypical codelength of the subsequence $x_{n}^{n+l-1}$ is
$$\begin{array}{cc} L_{A}\left(x_{n}^{n+l-1}\right) & =\left|S_{LZ}\left(x_{n}^{n+l-1}\right)\right|\left[\log\left|S_{LZ}\left(x_{n}^{n+l-1}\right)\right|+1\right]\\ \; & +\log^{*}\left(l\right)+\tau , \end{array}$$
where $\log^{*}\left(l\right)+\tau$ is an additive penalty for not knowing in advance the start and end points of the anomalous sequence [2,3], and $\log^{*}\left(l\right)=\log l+\log\log l+\ldots$ where the sum continues as long as the argument to the outer log is positive. Let $L_{A}^{{}^{\prime }}=L_{A}-\tau$. We propose the following atypicality criterion for detection of an anomalous subsequence:(6)$$\begin{array}{cc} ▵L\left(n\right) & =\max_{l}\left\{L_{T}\left(x_{n}^{n+l-1}\right)-L_{A}^{{}^{\prime }}\left(x_{n}^{n+l-1}\right)\right\}>\tau , \end{array}$$
where τ can be treated as an anomaly detection threshold. In practice, τ can be set to ensure a false positive constraint, e.g., using bootstrap estimation of the quantiles in the training data.

## 6. Experiment

In this section, we illustrate the proposed pattern dictionary anomaly detection on a synthetic time series, known as Mackey–Glass [77], as well as on a real-world time series of physiological signals. In both experiments, first, the real-valued samples are discretized using a uniform quantizer [78], and then, anomaly detection methods are applied.

### 6.1. Anomaly Detection in Mackey–Glass Time
Series

In this section, we illustrate the proposed anomaly detection method for the case of a chaotic Mackey–Glass (MG) time series that has an anomalous segment grafted into the middle of the sequence. MG time series are generated from a nonlinear time delay differential equation. The MG model was originally introduced to represent the appearance of complex dynamic in physiological control systems [77]. The nonlinear differential equation is of the form $\frac{dx\left(t\right)}{dt}=-ax\left(t\right)+\frac{bx\left(t-\delta \right)}{1+x^{10}\left(t-\delta \right)},\;t\geq 0$, where *a*, *b* and δ are constants. For the training data, we generated 3000 samples of the MG time series with a=0.2, b=0.1, and δ=17. For the test data, we normalized and embedded 500 samples of the MG time series with a=0.4, b=0.2, and δ=17 inside 1000 samples of a MG time series generated from the same source as the training data, resulting in a test sequence of length 1500. Figure 7 shows a realization of the training data and the test data.

The anomaly detection performance of our proposed pattern dictionary is evaluated. To illustrate the effect of the model parameter, i.e., the maximum depth $D_{max}$, on the detection and compression performance of the pattern dictionary, we run two experiments. First, we use a 30-fold cross-validation on the training data (resulting in 30 sequences of length 100) and calculate the number of distinct parsed phrases against $D_{max}$. Second, we train a pattern dictionary with various $D_{max}$ using the training data and then evaluate the sensitivity of detector of the anomalous subsequences in the test data using Equation (Equation 6) with τ=0. In this experiment, the detection sensitivity (true positive rate) is defined as the ratio of number of samples correctly identified as anomalous over the total number of anomalous samples. Figure 8 illustrates the result of both experiments. As seen, after some point, increasing $D_{max}$ has diminishing effect on both detection sensitivity and the number of distinct parsed phrases. Note that this behavior is to be expected as it was suggested by the bounds in (Equation 5).

Next, we compare anomaly detection performance of our proposed pattern dictionary methods, PDD and PDA, with the nearest neighbors-based similarity (NNS) technique [7], the compression-based dissimilarity measure (CDM) method [12,13,14], Ziv–Merhav method (ZM) [48], and the threshold Sequence Time-Delay Embedding (t-STIDE) technique [8,9,10,11]. In this experiment, a window of length 100 is slid over the test data and each method measures the *anomaly score* (as described below) of the current subsequence with respect to the training data. The anomaly is detected when the score exceeds a threshold, determined to ensure a specified false positive rate. In the following, we compute AUC (area under the curve) of the ROC (receiver operating characteristic) and Precision-Recall curves as performance measures. In the following, we provide details of the implementation.

Pattern Dictionary for Detection (PDD)

First, the training data are used to create a pattern dictionary with $D_{max}=40$, as described in Section 4. Then, for each subsequence $x^{100}$ (the sliding window of length 100) of the test data, the anomaly score is computed as the codelength $L\left(x^{100}\right)$ of Equation (Equation 2) described in Section 4.3.

Pattern Dictionary Based Atypicality (PDA)

Similar to PDD, first the training data are used to create a pattern dictionary with $D_{max}=40$, as described in Section 4. Then, for each subsequence $x^{100}$ of the test data, the anomaly score is the atypicality measure described in Section 5, i.e., $L_{T}\left(x^{100}\right)-L_{A}\left(x^{100}\right)$, the difference between the compression codelength of the test subsequence using typical encoder (pattern dictionary) and atypical encoder (LZ78).

Ziv–Merhav Method (ZM) [48]

In this method, a cross-parsing procedure is used in which for each subsequence $x^{100}$ of the test data, the anomaly score is computed as the number of the distinct phrases of $x^{100}$ with respect to the training data.

Nearest Neighbors-Based Similarity (NNS) [7] 

In this method, a list S of all the subsequence of length 100 (the length of the sliding window) of the training data is created. Then, for each subsequence $x^{100}$ of the test data, the distance between $x^{100}$ and all the subsequences in the list S is calculated. Finally, the anomaly score of $x^{100}$ is its distance to the nearest neighbor in the list S.

Compression-Based Dissimilarity Measure (CDM) [12,13,14] 

In this method, given the training data $x_{train}$, for each subsequence $x^{100}$ of the test data the anomaly score is
$$\begin{array}{cc} CDM(x_{train},x^{100}) & =\frac{L\left(C\left(x_{train},x^{100}\right)\right)}{L\left(x_{train}\right)+L\left(x^{100}\right)}, \end{array}$$
where $C\left(y,x\right)$ represents concatenation of sequences *y* and *z*, and $L\left(x\right)$ is the size of the compressed version of the sequence *x* using any standard compression algorithm. The CDM anomaly score is close to 1 if the two sequence are not related, and smaller than one if the sequences are related.

Threshold Sequence Time-Delay Embedding (t-STIDE) [8,9,10,11]

In this method, given l<100, for each sub-subsequence $x^{l}$ of the subsequence $x^{100}$ of the test data, the likelihood score of $x^{l}$ is the normalized frequency of its occurrence in the training data, and the anomaly score of $x^{100}$ is one minus the average likelihood score of all its sub-subsequences of length *l*. In this experiment, various values of *l* are tested and the best performance is reported.

We compare the detection performance of the aforementioned methods by generating 200 test data sequences with different anomaly segments (the anomalous MG segments have different initializations in each test dataset). The detection results of comparisons are reported in Table 2. As seen, our proposed PDD and PDA methods outperform the rest, with ZM and CDM coming in third place. The effect of alphabet size of the quantized data (the resolution parameter of the uniform quantizer [78]) on anomaly detection performance is summarized in Table 3. Table 3 shows that our proposed PDD and PDA methods outperform in all three cases of data resolution.

Since the parsing procedure of our proposed PD-based methods and the ZM method [48] are similar, it is of interest to compare the running time of these two methods. While the cross-parsing procedure of the ZM method was introduced as an on the fly process [48], we can also consider another implementation similar to our proposed PD by creating a codebook of all the subsequences of the training data prior to the parsing procedure. As such, in order to compare the running time of the dictionary/codebook creation and parsing procedure of our PD-based methods with the aforementioned two implementations of the ZM method, we use the same MG training data of length 3000, one test dataset of length 1500 while a sliding window of length 100 is slid over it for anomaly score calculation, and the PD-based method with $D_{max}=40$. Note that since a sliding window of length 100 over the test data is considered, for the codebook-based implementation of ZM, all the subsequences of the training data up to length 100 are extracted which make its codebook creation process significantly faster. Table 4 summarizes the running time comparison. As it can be seen, our PD-based method is faster in both dictionary/codebook creation and parsing process.

### 6.2. Infection Detection Using Physiological Signals

Finally, we apply the proposed pattern dictionary method to detect unusual patterns in physiological signals of two human subjects after exposure to a pathogen while only one of these subjects became symptomatically ill. The time series data were collected in a human viral challenge study that was performed in 2018 at the University of Virginia under a DARPA grant. Consented volunteers were recruited into this study following an IRB-approved protocol and the data was processed and analyzed at Duke University and the University of Michigan. The challenge study design and data collection protocols are described in [79]. Volunteers’ skin temperature and heart rate were recorded by a wearable device (Empatica E4) over three consecutive days before and five consecutive days after exposure to a strain of human Rhinovirus (RV) pathogen. During this period, the wearable time series were continuously recorded while biospecimens (viral load) were collected daily. The infection status can be clinically detected by biospecimen samples, but in practice, the collection process of these types of biosamples can be invasive and costly. As such, here, we apply the proposed anomaly detection framework to the measured two-dimensional heart rate and temperature time series to detect unusual patterns after exposure with respect to the normal (healthy) baseline patterns.

In the preprocessing phase, we followed the wearable data preprocessing procedure described in [80]. Specifically, we first downsample the time series to one sample per minute by averaging. Then, we apply an outlier detection procedure to remove technical noise, e.g., sensor contact loss. After preprocessing, the two-dimensional space of temperature and heart rate time series is discretized using a two-dimensional uniform quantizer [78] with step size of 5 for heart rate and 0.5 for temperature, resulting in one-dimensional discrete sequence data. The first three days of data are used as the training data, and the PDA methods with maximum depth $D_{max}=30$ are used to learn the patterns in the training data. In order to detect anomalous patterns of the test data (the last five days), we used the result of Section 5.3 and the atypicality criterion of Equation (Equation 6), which requires choosing the threshold τ. While this threshold can be chosen freely, we selected it using cross-validation on the training data. Leave-one-out cross-validation over the training data generates an empirical null distribution of the PDA anomaly score function $L_{T}-L_{A}$. The threshold τ was chosen as the upper 99% quantile of this distribution. Figure 9 illustrates the result of anomaly detection on one subject who became infected as measured by viral shedding as shown in Figure 9C. All the anomalous patterns occur when the subject was shedding the virus. Figure 10 also depicts the result of anomaly detection on one subject who had a mild infection with a low level of viral shedding, as shown in Figure 10C. Note that in this case, no anomalous patterns were detected.

## 7. Conclusions

In this paper, we have developed a universal nonparametric model-free anomaly detection method for time series and sequence data using a pattern dictionary. We proved that using a multi-level dictionary that separates the patterns by their depth results in a shorter average indexing codelength in comparison to a uni-level dictionary that uses a uniform indexing approach. We illustrated that the proposed pattern dictionary method can be used as a stand-alone anomaly detector, or integrated with Tree-Structured Lempel–Ziv (LZ78) and incorporated into an atypicality framework. We developed novel non-asymptotic lower and upper bounds of the LZ78 parser and demonstrated that the non-asymptotic upper bound on the number of distinct phrases resulting from LZ78-parsing of an $\left|X\right|$-ary sequence can be explicitly derived in terms of the Lambert W function, an important theoretical result that is not trivial. We showed that the achieved non-asymptotic bounds on LZ78 and pattern dictionary determine the range of the anomaly score and the anomaly detection threshold. We also presented an empirical study in which the pattern dictionary approach is used to detect anomalies in physiological time series. In the future work, we will investigate the generalization of the context tree weighting methods to the general discrete case, using the pattern dictionary since the pattern dictionary handles sparsity well and is computationally less expensive when the alphabet size is large.

## Figures and Tables

**Figure 1 entropy-24-01095-f001:**
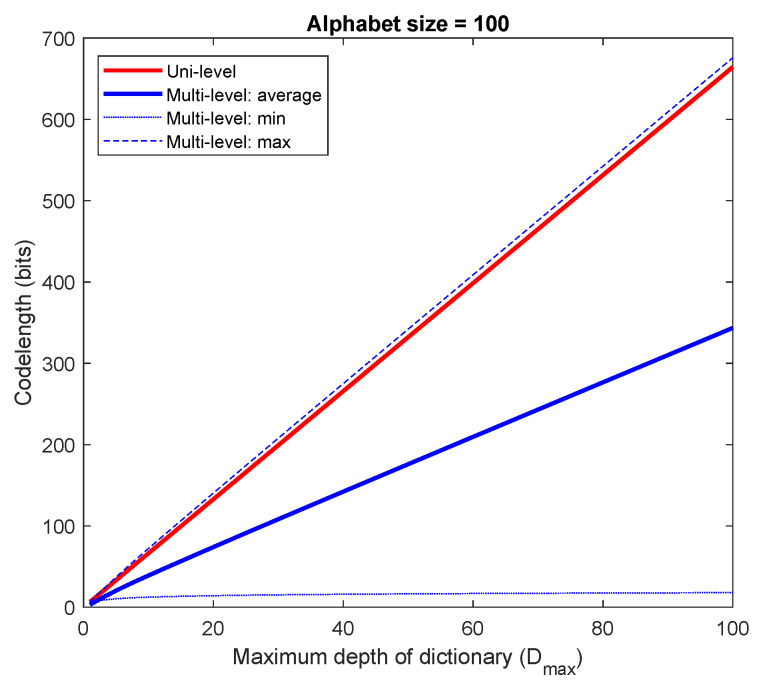
Comparison of indexing codelength between a uni-level dictionary and a multi-level dictionary (fixed alphabet size $\left|X\right|=100$).

**Figure 2 entropy-24-01095-f002:**
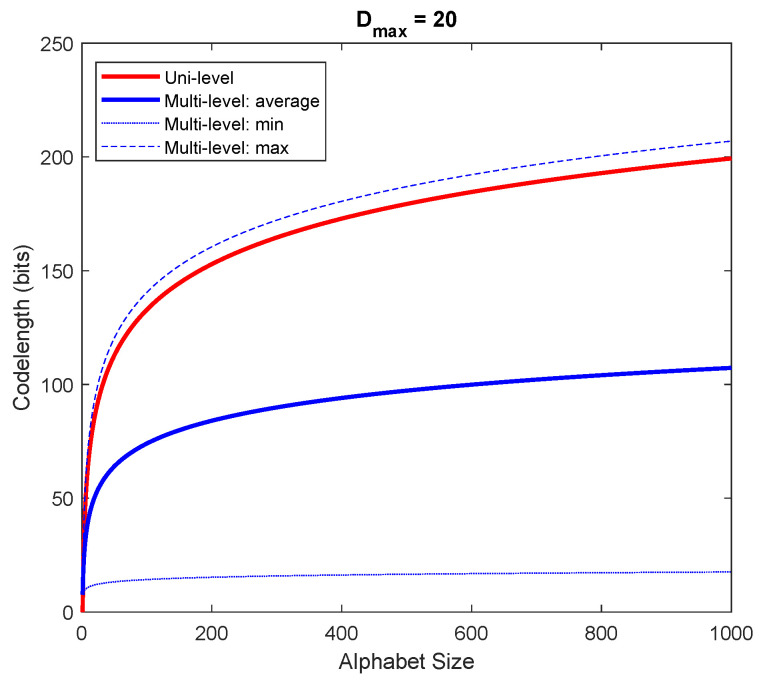
Comparison of indexing codelength between a uni-level dictionary and a multi-level dictionary (fixed $D_{max}=20$).

**Figure 3 entropy-24-01095-f003:**
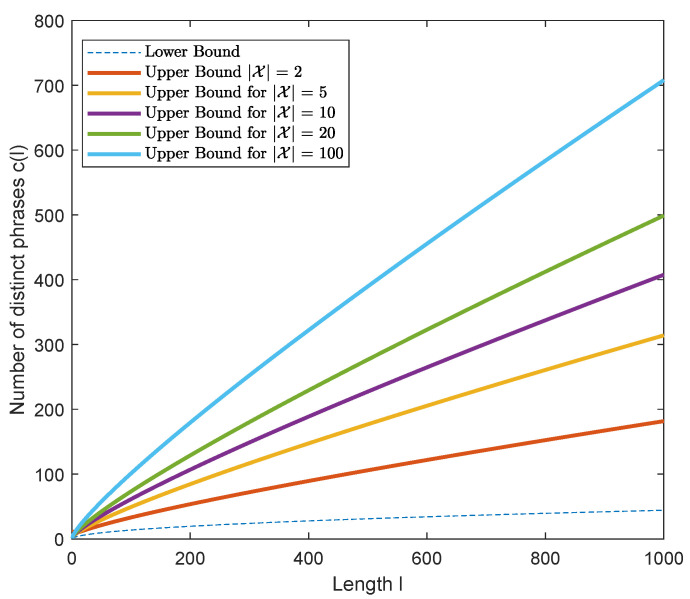
Plot of the lower and upper bounds of Theorem 2 on the number of distinct phrases resulting from LZ78-parsing of an $\left|X\right|$-ary sequence of length *l*.

**Figure 4 entropy-24-01095-f004:**
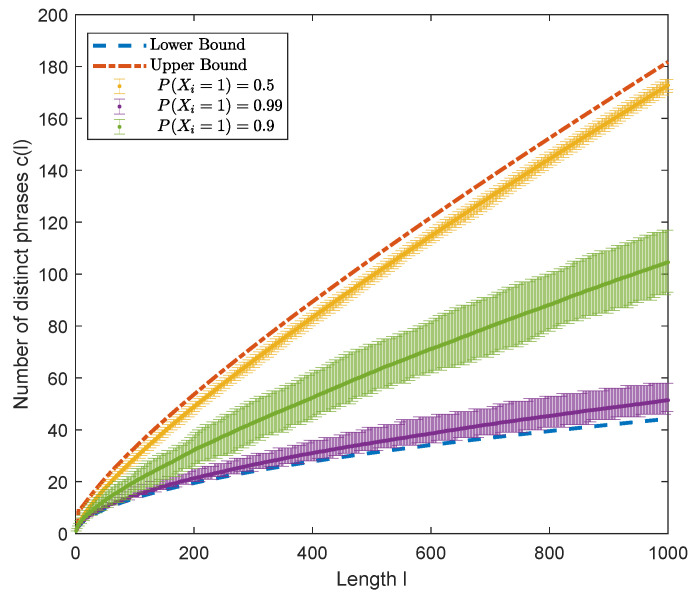
Simulation results compared to the lower and upper bounds of Theorem 2 on the number of distinct phrases resulting from LZ78-parsing of binary sequences of length *l* generated by sources with three different source probabilities $P\left(X=1\right)$. For every $P\left(X=1\right)$, one thousand binary sequences of length *l* are generated. Error bars represent the maximum, minimum, and average number of distinct phrases.

**Figure 5 entropy-24-01095-f005:**
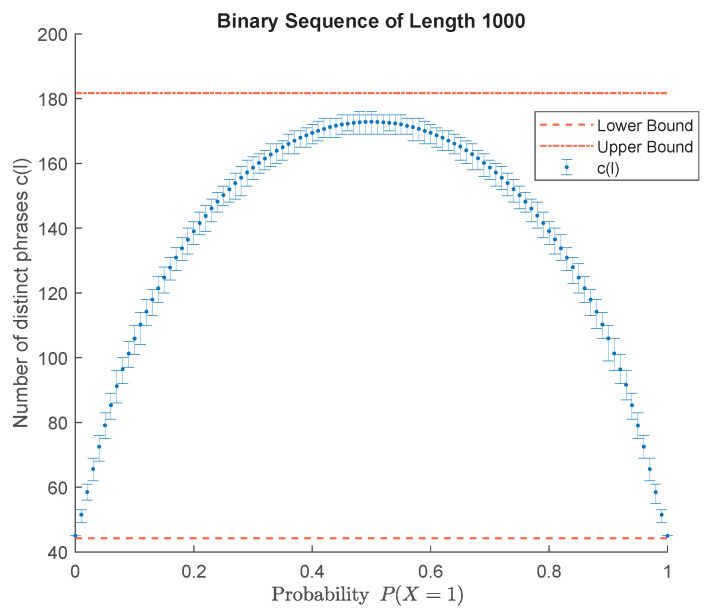
Similar to Figure 4, the number of distinct phrases resulting from LZ78-parsing of binary sequences of fixed length l=1000 varies over the source probability parameter $P\left(X=1\right)$. For every $P\left(X=1\right)$, one thousand binary sequences of length *l* are generated. Error bars represent the maximum, minimum, and average number of distinct phrases.

**Figure 6 entropy-24-01095-f006:**
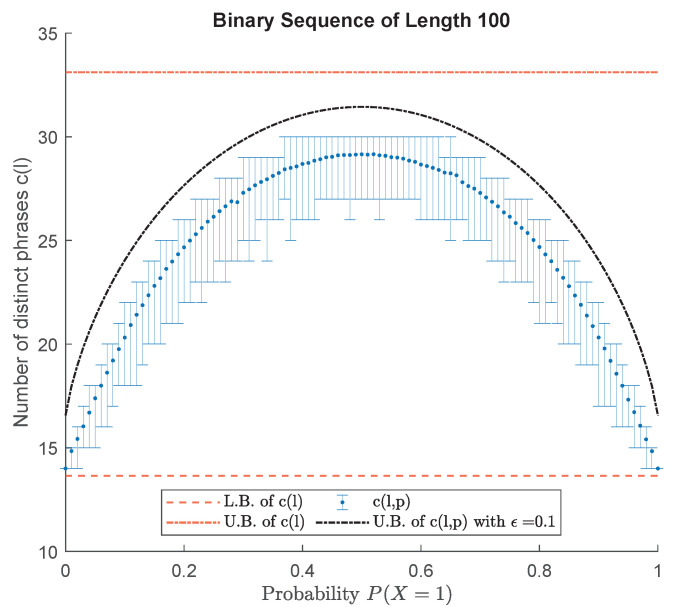
Simulation of the probability-dependent upper bound $c\left(l,p\right)$ for binary sequences of fixed length l=100 with various probability parameters $P\left(X=1\right)$. For every $P\left(X=1\right)$, one thousand binary sequences of length *l* are generated. Error bars represent the maximum, minimum, and average number of distinct phrases.

**Figure 7 entropy-24-01095-f007:**
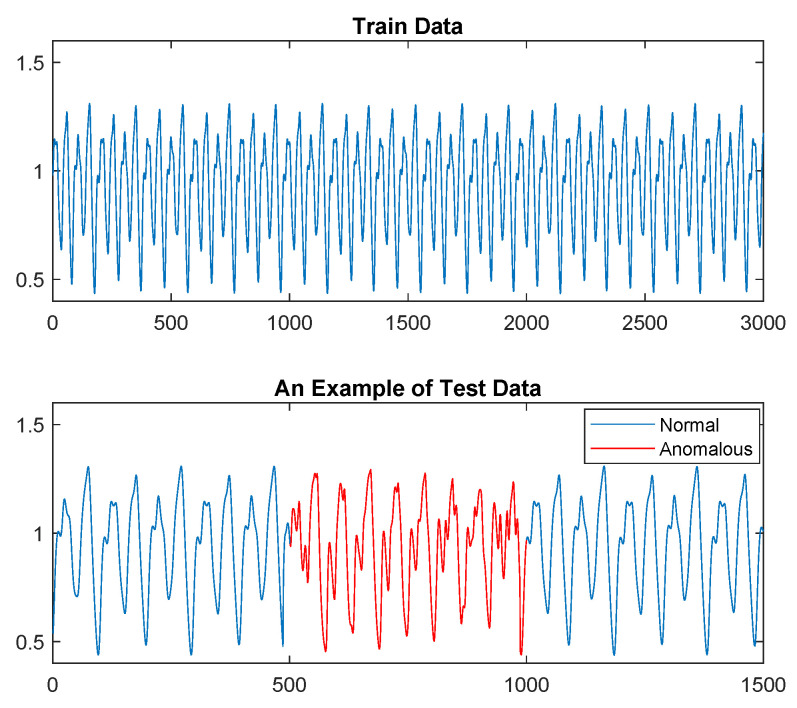
Mackey–Glass time series: the training data (**top**) and an example of the test data (**bottom**) in which samples in $\left[501,1000\right]$ are anomalous (shown in red).

**Figure 8 entropy-24-01095-f008:**
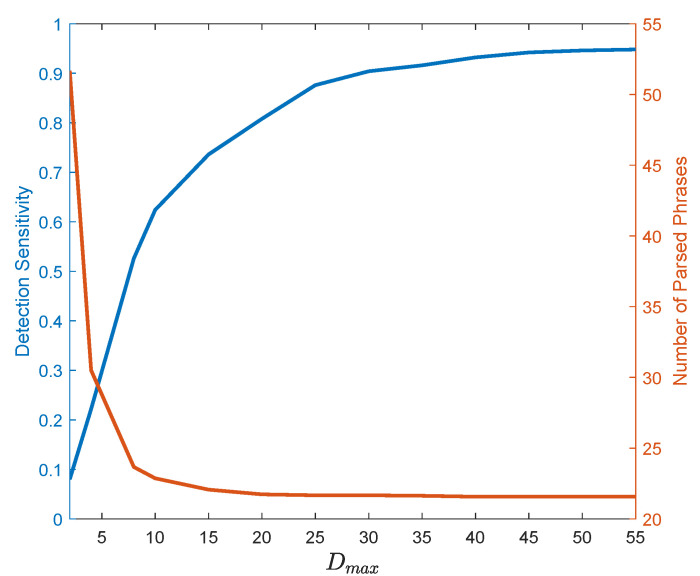
The effect of maximum dictionary depth $D_{max}$ on parsing and detection sensitivity (true positive rate) of the Mackey–Glass time series presented in Figure 7.

**Figure 9 entropy-24-01095-f009:**
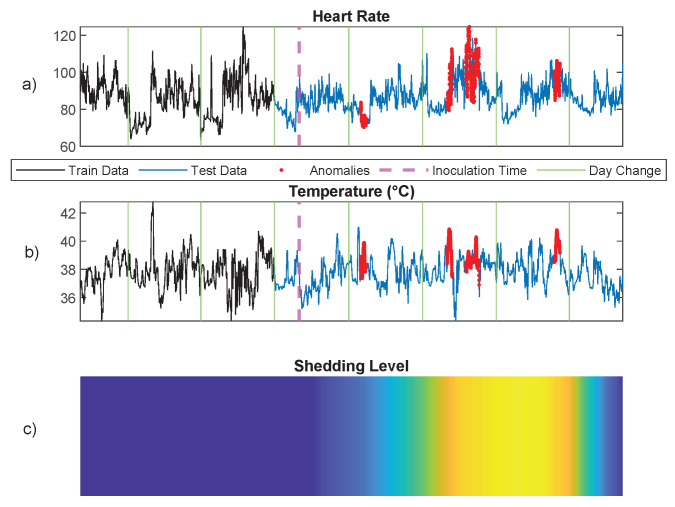
Anomaly detection using the proposed PDA method for a subject based on heart rate and temperature data collected from a wearable wrist sensor. Anomalies are shown in red in (**a**,**b**). (**c**) shows the subject’s infection level.

**Figure 10 entropy-24-01095-f010:**
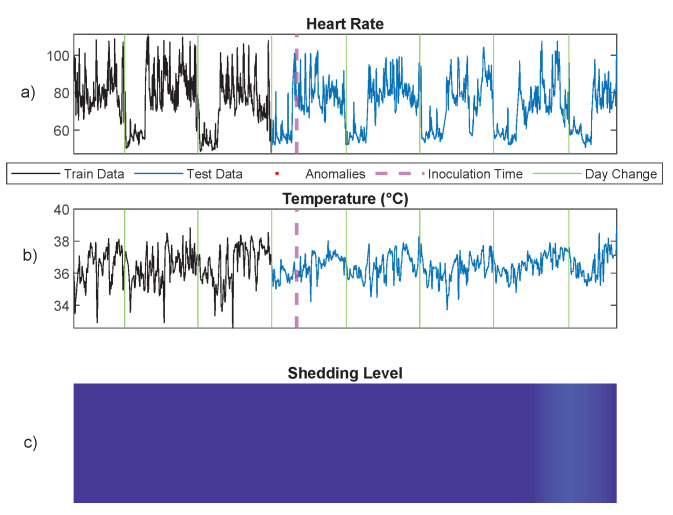
Anomaly detection using the proposed PDA method for a subject who had a mild infection with low level of viral shedding based on heart rate and temperature data collected from a wearable wrist sensor. Note that no anomaly has been detected: (**a**) heart rate, (**b**) temperature, and (**c**) infection level.

**Table 1 entropy-24-01095-t001:** Filling (training) the dictionary (of maximum depth $D_{max}=3$) with the patterns in the training sequence ABACADABBACCADDABABACADAB.

Depth 1			Depth 2			Depth 3		
$x_{1}^{d}$	Pr$\left(x_{1}^{d}\right)$	$L_{D}^{\left(1\right)}\left(x_{1}^{d}\right)$	$x_{1}^{d}$	Pr$\left(x_{1}^{d}\right)$	$L_{D}^{\left(2\right)}\left(x_{1}^{d}\right)$	$x_{1}^{d}$	Pr$\left(x_{1}^{d}\right)$	$L_{D}^{\left(3\right)}\left(x_{1}^{d}\right)$
A	0.44	1	AB	0.2083	2	ABA	0.1304	3
B	0.24	2	BA	0.1667	3	BAC	0.1304	3
C	0.16	3	AC	0.1250	3	CAD	0.1304	3
D	0.16	3	CA	0.1250	3	DAB	0.1304	3
			AD	0.1250	3	ACA	0.0870	4
			DA	0.1250	3	ADA	0.0870	4
			BB	0.0417	4	ABB	0.0435	4
			CC	0.0417	5	BBA	0.0435	4
			DD	0.0417	5	ACC	0.0435	4
						CCA	0.0435	4
						ADD	0.0435	4
						DDA	0.0435	5
						BAB	0.0435	5

**Table 2 entropy-24-01095-t002:** Comparison of anomaly detection methods (μ±σ representation is used where μ is the mean and σ is the standard deviation). The proposed PDA method attains overall best performance (bold entries of table).

	ROC AUC	PR AUC
PDA	0.963±0.009	0.909±0.044
PDD	0.959±0.009	0.907±0.044
ZM	0.959±0.009	0.895±0.049
CDM	0.957±0.012	0.907±0.057
NNS	0.920±0.021	0.777±0.091
t-STIDE	0.897±0.013	0.857±0.044

**Table 3 entropy-24-01095-t003:** Comparison of anomaly detection methods for different cases of data resolutions: high resolution corresponds to an alphabet size of 90, medium resolution corresponds to an alphabet size of 45, and low resolution corresponds to an alphabet size of 10. In this table, μ±σ representation is used where μ is the mean and σ is the standard deviation. The proposed PDA method achieves overall best performance (bold entries of table).

	Resolution	PDA	PDD	ZM	CDM	NNS	t-STIDE
ROC AUC	Low	**0.948** ±**0.011**	0.930 ±0.013	0.943 ±0.014	0.787 ±0.017	0.901 ±0.027	0.725 ±0.025
	Medium	**0.955** ±**0.010**	0.943 ±0.011	0.954 ±0.011	0.940 ±0.014	0.918 ±0.022	0.881 ±0.017
	High	**0.963** ±**0.009**	0.959 ±0.009	0.959 ±0.009	0.957 ±0.012	0.920 ±0.021	0.897 ±0.013
PR AUC	Low	**0.876** ±**0.050**	0.871 ±0.052	0.826 ±0.071	0.669 ±0.067	0.719 ±0.098	0.678 ±0.067
	Medium	**0.885** ±**0.046**	0.882 ±0.047	0.881 ±0.053	0.880 ±0.060	0.777 ±0.093	0.828 ±0.050
	High	**0.909** ±**0.044**	0.907 ±0.044	0.895 ± 0.044	0.907 ±0.057	0.777 ±0.091	0.857 ±0.044

**Table 4 entropy-24-01095-t004:** Comparison of running time (in second) of PD-based method and two implementations of the ZM method for different cases of data resolutions: high resolution corresponds to an alphabet size of 90, medium resolution corresponds to an alphabet size of 45, and low resolution corresponds to an alphabet size of 10. This experiment is performed on a Hansung laptop with 2.60 GHz CPU, 500 GB of SSD, and 16 GB of RAM using MATLAB R2021a. The proposed PD-based method has fastest run time overall (bold entries in table).

	Resolution	PD-Based	ZM-Codebook	ZM
dictionary generation	Low	**6.80**	29.98	N/A
	Medium	**13.12**	39.01	N/A
	High	**15.46**	40.80	N/A
parsing procedure	Low	**6.07**	9.23	142.77
	Medium	**10.81**	11.10	433.55
	High	**14.83**	16.70	670.18

## Data Availability

The experimental data used in Section 6.2 will be made available upon request.

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
