# Peer review of "A Pattern Dictionary Method for Anomaly Detection"

_entropy, 2022, doi:10.3390/e24081095_

Round 1

Reviewer 1 Report

This paper reports two anomaly detection methods based on lossless compression.

I have some minor comments that will sound major, largely because I am missing background information-- but other readers may be in the same boat!

First, in Section 5, this anomaly detection method will identify things as anomalous when a typical and an atypical encoder disagree.  This means you will pick up on some things that most people would call anomalous as not anomalous, e.g. when entropy rate changes.  It would be great to explain why this is desirable.

Second, in Figs 4-6, you have a quick way of communicating the validity of your proofs (a strong point in the paper, I think, though I did not check them carefully), but it would be great if you had Figs 4-6 repeated for more complicated input than binary valued i.i.d. process.  Apologies if I misunderstood the input for the figures (but then please explain the input in the captions!)

Finally, in Section 6, you identify a particular snippet of the time series generated by the Mackey-Glass equation as anomalous.  I would not, from a stochastic processes point of view, see it in that way.  And in Table 2, I'm not sure where ground truth comes from.  Elaborating on this would be great.  Why is that snippet anomalous intuitively, and where does ground truth come from?

I am happy to recommend this strong paper for acceptance once these issues have been taken care of.

Author Response

We would like to thank the reviewer for the constructive comments. A rebuttal is attached as a PDF file.

Reviewer 2 Report

The paper developes a universal nonparametric model-free anomaly detection method for time series and sequence data using a pattern dictionary. It proves that using a multi-level dictionary that separates the patterns by their depth results in a shorter average indexing code length in comparison to a uni-level dictionary that uses a uniform indexing approach.

In general, the paper is well written and shows some contributions in the field of anomaly detection. My recommendation is to accept this paper in its present form.

Author Response

(The authors gave the same response as above.)
